# Phytochemicals from *Vanda bensonii* and Their Bioactivities to Inhibit Growth and Metastasis of Non-Small Cell Lung Cancer Cells

**DOI:** 10.3390/molecules27227902

**Published:** 2022-11-15

**Authors:** Tajudeen O. Jimoh, Narawat Nuamnaichati, Rungroch Sungthong, Chaisak Chansriniyom, Pithi Chanvorachote, Kittisak Likhitwitayawuid, Chatchai Chaotham, Boonchoo Sritularak

**Affiliations:** 1Pharmaceutical Sciences and Technology Program, Faculty of Pharmaceutical Sciences, Chulalongkorn University, Bangkok 10330, Thailand; 2Department of Biochemistry, Faculty of Health Sciences, Islamic University in Uganda, Kampala P.O. Box 7689, Uganda; 3Department of Biochemistry and Microbiology, Faculty of Pharmaceutical Sciences, Chulalongkorn University, Bangkok 10330, Thailand; 4Department of Pharmacognosy and Pharmaceutical Botany, Faculty of Pharmaceutical Sciences, Chulalongkorn University, Bangkok 10330, Thailand; 5Natural Products and Nanoparticles Research Unit, Chulalongkorn University, Bangkok 10330, Thailand; 6Department of Pharmacology and Physiology, Faculty of Pharmaceutical Sciences, Chulalongkorn University, Bangkok 10330, Thailand; 7Center of Excellence in Cancer Cell and Molecular Biology, Faculty of Pharmaceutical Sciences, Chulalongkorn University, Bangkok 10330, Thailand; 8Center of Excellence in Natural Products for Ageing and Chronic Diseases, Faculty of Pharmaceutical Sciences, Chulalongkorn University, Bangkok 10330, Thailand

**Keywords:** *Vanda bensonii*, phytochemicals, lung cancer, anticancer, metastasis, cytotoxicity, cell proliferation, cell migration, anchorage-independent growth

## Abstract

The most prevalent lung cancer is non-small cell lung cancer (NSCLC). This lung cancer type often develops other organ-specific metastases that are critical burdens in the treatment process. Orchid species in the genus *Vanda* have shown their potential in folkloric medication of diverse diseases but not all its species have been investigated, and little is known about their anticancer activities against NSCLC. Here, we firstly profiled the specialized metabolites of *Vanda bensonii* and examined their capability to inhibit growth and metastasis of NSCLC using NCI-H460 cells as a study model. Four phytochemicals, including phloretic acid methyl ester (**1**), cymbinodin-A (**2**), ephemeranthoquinone B (**3**), and protocatechuic acid (**4**), were isolated from the whole plant methanolic extract of *V. bensonii*. The most distinguished cytotoxic effect on NCI-H460 cells was observed in the treatments with crude methanolic extract and compound **2** with the half maximal inhibitory concentrations of 40.39 μg mL^−1^ and 50.82 μM, respectively. At non-cytotoxic doses (10 μg mL^−1^ or 10 μM), only compound **1** could significantly limit NCI-H460 cell proliferation when treated for 48 h, while others excluding compound **4** showed significant reduction in cell proliferation after treating for 72 h. Compound **1** also significantly decreased the migration rate of NCI-H460 cells examined through a wound-healing assay. Additionally, the crude extract and compound **1** strongly affected survival and growth of NCI-H460 cells under anchorage-independent conditions. Our findings proved that natural products from *V. bensonii* could be promising candidates for the future pharmacotherapy of NSCLC.

## 1. Introduction

The genus *Vanda* in the family Orchidaceae is among the five most horticulturally important orchid genera in the world, which contains over 70 member species found across Southern and Southeastern Asia [1]. They are mostly epiphytes, while some can have lithophytic and monopodial growth features. Some of them are employed as unrefined drugs with a domestic cognomen “Rasna” in Ayurvedic medicine [2]. Their folkloric medical uses are also perceived across Nepal, Thailand, India, Bangladesh, and many other countries in South Asia [3]. Several *Vanda* species (e.g., *Vanda testacea*, *Vanda cristata*, *Vanda parviflora*, *Vanda spathulata*, and *Vanda roxburghii*) have been unveiled for their medical applications in the treatment of diverse illnesses [2,3]. Although *Vanda* species have shown their recognized healing capabilities, further in-depth studies on their elucidated bioactive molecules are essential to extend their pharmaceutical and medical implementations.

In Indian folkloric medicine, the juice from *Vanda coerulea* leaves can improve indigestion and diarrhea [4], while its flower decoction is applied as a tonic and appetizer [5]. Similarly, the paste made from the roots or whole plants of *V. cristata* is used to treat dislocated bones and heal varieties of skin wounds [6], while its leaf extracts exhibit curing capacities for cough, bronchitis, weakness, and tonsillitis [4]. Interestingly, applying different forms of *Vanda* crude mixtures containing varying bioactive compounds for healing diseases is a promising and straightforward medication to support human well-being. Some further studies proved that *Vanda* plants are the rich sources of bibenzyls, phenanthrenes, anthocyanins, and simple phenolic compounds, which exert some promising pharmaceutical properties, e.g., hepatoprotective, antioxidant, anticonvulsant, anti-inflammatory, and anti-aging [2,3,7,8]. Only few reviews reported anticancer activities of *V. parviflora* [2,3], but the relevant research evidence is rarely available. To our knowledge, *Vanda* phytochemicals and their roles in cancer therapy are yet scarce.

Based on the global prevalence and mortality rate, 2,206,771 (11.4%) new cases and 1,796,144 (18.0%) new deaths were recorded for patients with lung cancer [9]. Among all lung cancer cases, patients with non-small cell lung cancer (NSCLC) are the majority (approximately over 80% [10]) compared to those with small cell lung cancer (SCLC). Unlike SCLC that is faster in disease progression and greater severity, slower progress in NSCLC results in the late-phase detection of the disease and allows the emergence of secondary malignant growths at other sites of the body, so-called “metastasis”. NSCLC patients have been prognosticated to have bone, brain, and many other organ-specific metastases [11,12]. Metastasis involves cancer cell migration, leading to survival and growth under anchorage-independent conditions. This aggressive feature of cancer cells is one of the major impediments in cancer therapy, which still requires advanced therapeutic approaches and more effective medicines with fewer or no side effects. Hence, exploiting pharmaceutical benefits of natural products to inhibit growth and metastasis of NSCLC cells is yet a scientific challenge in the development of alternative treatments to defeat lung cancer.

Among the notable *Vanda* species in Thailand [13], *Vanda bensonii* Batem., with Thai name “Sam Poi Chomphu,” is an economically important orchid species due to its beautifully unique flower features (Figure 1). This *Vanda* species is still unknown for its chemical constituents and their pharmaceutical potential. Unveiling the phytochemical profile of this *Vanda* species would benefit orchid chemotaxonomy and enhance its ethnopharmacology. In this study, we profiled the specialized metabolites of *V*. *bensonii* cultivated in Thailand and elucidated their capability to inhibit growth and metastasis of NSCLC cells using the human NCI-H460 cells as a studied model. These NSCLC cells were chosen based on their origin from pleural fluid, which represent high metastatic features, including capability to grow under anchorage-independent conditions [14].

## 2. Results

### 2.1. Key Phytochemicals Isolated from V. bensonii

From the methanolic extract prepared from the whole *V*. *bensonii* plant, a variety of specialized metabolites, including phloretic acid methyl ester (**1**), cymbinodin-A (**2**), ephemeranthoquinone B (**3**), and protocatechuic acid (**4**), were isolated (Figure 1 and Table 1). The chemical structures of each compound were characterized by extensive spectroscopic studies, including 1D and 2D-NMR as well as HR-ESI-MS experiments.

Phloretic acid methyl ester (**1**): White powder; HR-ESI-MS: [M+Na]^+^ at *m*/*z* 203.0674 (calcd. for 203.0684, C_10_H_12_O_3_Na); ^1^H NMR (400 MHz, acetone-*d*_6_) δ: 8.13 (1H, s, HO-4), 7.04 (2H, d, *J* = 8.4 Hz, H-2, H-6), 6.74 (2H, d, *J* = 8.4 Hz, H-3, H-5), 3.60 (3H, s, MeO-9), 2.80 (2H, t, *J* = 7.6, H_2_-7), 2.54 (2H, t, *J* = 7.6, H_2_-8); ^13^C NMR (100 MHz, acetone-*d*_6_) δ: 173.5 (C-9), 156.7 (C-4), 132.5 (C-1), 130.2 (C-2, C-6), 116.1 (C-3, C-5), 51.6 (MeO-9), 36.6 (C-8), 30.6 (C-7) (Figure 1).

Cymbinodin-A (**2**): Red amorphous solid; HR-ESI-MS: [M+H]^+^ at *m*/*z* 255.0650 (calcd. for 255.0657, C_15_H_11_O_4_); ^1^H NMR (400 MHz, acetone-*d*_6_) δ: 12.35 (1H, s, HO-5), 8.34 (1H, d, *J* = 8.8 Hz, H-9), 8.14 (1H, d, *J* = 8.8 Hz, H-10), 7.63 (1H, dd, *J* = 7.6, 8.0 Hz, H-7), 7.55 (1H, dd, *J* = 8.0, 1.6 Hz, H-8), 7.16 (1H, dd, *J* = 7.6, 1.6 Hz, H-6), 6.50 (1H, s, H-3), 4.05 (3H, s, MeO-2); ^13^C NMR (100 MHz, acetone-*d*_6_) δ: 193.5 (C-4), 180.6 (C-1), 160.5 (C-2), 156.7 (C-5), 140.0 (C-8a), 138.1 (C-9), 133.8 (C-10a), 131.6 (C-7), 130.9 (C-4a), 122.6 (C-10), 121.8 (C-8), 117.8 (C-6), 112.4 (C-3), 57.3 (MeO-2) (Figure 1).

Ephemeranthoquinone B (**3**): Red amorphous solid; HR-ESI-MS: [M+Na]^+^ at *m*/*z* 279.0633 (calcd. for 279.0633, C_15_H_12_O_4_Na); ^1^H NMR (400 MHz, acetone-*d*_6_) δ: 9.65 (1H, s, HO-5), 7.21 (1H, dd, *J* = 7.6, 8.0 Hz, H-7), 6.81 (2H, br d, *J* = 8.0 Hz, H-6, H-8), 6.18 (1H, s, H-3), 3.94 (3H, s, MeO-2), 2.72 (2H, m, H_2_-9), 2.58 (2H, m, H_2_-10); ^13^C NMR (100 MHz, acetone-*d*_6_) δ: 191.5 (C-4), 181.3 (C-1), 160.0 (C-2), 156.4 (C-5), 143.8 (C-10a), 142.1 (C-8a), 139.8 (C-4a), 132.6 (C-7), 120.6 (C-8), 118.9 (C-6), 118.6 (C-4b), 108.9 (C-3), 57.2 (MeO-2), 29.1 (C-9), 21.9 (C-10) (Figure 1).

Protocatechuic acid (**4**): White powder; HR-ESI-MS: [M−H]^−^ at *m*/*z* 153.0192 (calcd. for 153.0187, C_7_H_5_O_4_); ^1^H NMR (400 MHz, methanol-*d*_4_) δ: 6.74 (1H, d, *J* = 8.4 Hz, H-5), 7.38 (1H, dd, *J* = 8.4, 2.0 Hz, H-6), 7.42 (1H, d, *J* = 2.0 Hz, H-2); ^13^C NMR (100 MHz, methanol-*d*_4_) δ: 173.9 (COOH), 150.1 (C-4), 145.7 (C-3), 127.9 (C-1), 123.5 (C-6), 117.9 (C-2), 115.5 (C-5) (Figure 1).

### 2.2. The Orchid Metabolites Exhibit a Broad Spectrum of Anticancer Activities

Cytotoxic effects of the crude methanolic extract and each isolated compound from *V. bensonii* varied depending upon the tested compounds and concentrations (Figure 2 and Table 1). The crude methanolic extract at 25 μg mL^−1^ started to reduce NCI-H460 cell viability significantly (Figure 2a), while cymbinodin-A (**2**) at 25 μM started to show cytotoxic effects (Figure 2e). The absolute half maximal inhibitory concentration (absolute IC_50_) value (Table 1) computed from the sigmoidal dose-response plots (Appendix A) was 40.39 μg mL^−1^ for the crude methanolic extract and 50.82 μM for cymbinodin-A (**2**). Phloretic acid methyl ester (**1**) and ephemeranthoquinone B (**3**) showed cytotoxic effects on NCI-H460 cells starting from their concentration at 50 μM (Figure 2c,g), while protocatechuic acid (**4**) did not exhibit cytotoxicity up to the maximum tested concentration of 100 μM (Figure 2i). The absolute IC_50_ value for ephemeranthoquinone B (**3**) (64.22 μM) was greater than that of phloretic acid methyl ester (**1**) (>100 μM) and protocatechuic acid (**4**) (>100 μM) (Table 1 and Appendix A). The difference in cytotoxic profiles for each tested compound correlated with a dual Hoechst 33342/propidium iodide nuclear staining assay (Figure 2b,d,f,h,j), in which dead NCI-H460 cells were only observed when tested with cytotoxic concentrations. With these findings, the concentrations up to 10 μg mL^−1^ or 10 μM were defined as the non-cytotoxic doses for all tested compounds.

*V. bensonii* metabolites affected NCI-H460 cell proliferation differently (Figure 3 and Table 1). Every tested compound did not show an inhibitory effect on NCI-H460 cell proliferation when testing for 24 h (Figure 3). Except for protocatechuic acid (**4**), other tested *V. bensonii* metabolites could decrease NCI-H460 cell proliferation after exposure for 48 h, although a significant reduction 0.80-fold lower than the vehicle control was only observed with the phloretic acid methyl ester (**1**) treatment (Figure 3 and Table 1). The anti-proliferation degree of NCI-H460 cells increased when extending the incubation time with *V. bensonii* metabolites to 72 h (Figure 3 and Table 1).

Migration and anchorage-independent growth of NCI-H460 cells in the presence of *V*. *bensonii* metabolites were assessed to unveil the impacts of these orchid metabolites on NSCLC cell metastasis. A wound-healing assay was conducted to investigate how *V*. *bensonii* compounds hampered NCI-H460 cell migration (Figure 4). The wound area decreased when the incubation time increased (Figure 4a). Changes in relative wound areas at 12 and 24 h to those measured at 0 h were not significantly different when compared among different *V*. *bensonii* metabolites (Figure 4b). When considering the wound-healing rate of NCI-H460 cells treated with each *V*. *bensonii* compound, the treatment with phloretic acid methyl ester (**1**) offered the maximum reduction (0.14 ± 0.01 h^−1^) followed by the crude methanolic extract and cymbinodin-A (**2**) (Figure 4c and Table 1).

The capability of NCI-H460 cells to grow without anchorage in the presence of *V*. *bensonii* metabolites was assessed using a soft agar colony formation assay (Figure 5 and Table 1). The number and size of NCI-H460 colonies formed in the soft agar matrix represented the capability of NCI-H460 cells to survive and grow under anchorage-independent conditions, respectively. Varying sizes of NCI-H460 colonies formed in the soft agar after exposure to different tested compounds for 7 days were visualized and examined (Figure 5a). Except for ephemeranthoquinone B (**3**), other *V*. *bensonii* metabolites significantly decreased the number of NCI-H460 colonies, while the maximum reduction in NCI-H460 cell survival (0.43-fold lower than that of the vehicle control) was found in the treatment with phloretic acid methyl ester (**1**) (Figure 5b and Table 1). A similar trend of results was observed with the sizes of NCI-H460 colonies exposed to *V*. *bensonii* metabolites, where the maximum size reduction 0.20-fold lower than that of the vehicle control was found with the phloretic acid methyl ester (**1**) treatment (Figure 5c and Table 1). However, cymbinodin-A (**2**) only significantly affected the number of NCI-H460 colonies but did not affect the colony sizes (Figure 5b,c). Protocatechuic acid (**4**) showed greater effect on the survival of NCI-H460 cells than their growth under the anchorage-independent conditions (Figure 5b,c).

## 3. Discussion

All four isolated compounds (**1**–**4**) from *V*. *bensonii* have been reported to be isolated from a variety of sources [15,16,17,18,19,20,21,22,23,24,25,26,27,28], but the information regarding their roles in anticancer activities, especially for antimetastatic roles, is still limited. To develop secondary malignant sites, cancer cells migrate and invade into vascular and lymphatic systems [11,12,14]. These migrating cancer cells evolve anoikis resistance, which enables them to survive and grow without anchorage and to form new tumor sites after extravasation [29]. This migration capability and anchorage-independent survival and growth in cancer metastasis are recognized aggressive characteristics of cancerous cells, which becomes a critical burden in cancer therapy. Compounds that can inhibit migration, growth, and survival of cancer cells under both anchorage-dependent and anchorage-independent conditions would be promising pharmaceutical products to cure cancers. Our findings proved that crude methanolic extract and phloretic acid methyl ester (**1**) derived from *V*. *bensonii* exhibited the most distinguished inhibitory activities against growth and metastasis of human NSCLC cells.

Phloretic acid methyl ester (**1**) was found to be a product of phloretic acid transformation by human intestinal bacteria, while the precursor of such biotransformation is phlorizin, a common phenolic compound of apple trees [15]. Phloretic acid is also a predominant phenolic compound found in *Cucurbita pepo* (field pumpkin) flowers, and its methyl ester is a recognized insect attractant [16]. To our knowledge, pharmaceutical roles of phloretic acid methyl ester (**1**), especially for its anticancer activities, have yet been unknown. Our results firstly proved that this *V*. *bensonii* metabolite can lower viability, proliferation, migration, and anchorage-independent survival and growth of NCI-H460 cells. In comparison with the crude methanolic extract derived from *V*. *bensonii*, it is highly possible that phloretic acid methyl ester (**1**) is the key anticancer agent found in this plant. Strong anticancer activities of *V*. *bensonii*-derived crude methanolic extract would offer the feasibility of using this pool of bioactive compounds in cancer treatment. However, additional studies are yet required to assess its toxicity and healing capacity.

Cymbinodin-A (**2**) is a phenanthraquinone firstly found in an aloe-leafed orchid *Cymbidium aloifolium* [17] and recently found in another *Cymbidium* species—*Cymbidium finlaysonianum*—with a proposed revised chemical structure [18]. Li et al. [19] also found this compound in *Gastrodia elata*, an herbal composition of a Chinese “Tianshu” formula for migraine therapy. The compound possesses diverse bioactivities, including anticancer [18] and anti-collagenase properties [20]. The anticancer activity of this phytochemical was assessed with human small cell lung cancer (NCI-H187) cells, and its IC_50_ value was 3.73 μM, which was greater and weaker than that of the positive controls, ellipticine (17.09 μM) and doxorubicin (0.24 μM), respectively [18]. *V*. *bensonii* also contains cymbinodin-A (**2**) and its closely related molecule, ephemeranthoquinone B (**3**) (Figure 1). Our study has proved that *V*. *bensonii* cymbinodin-A (**2**) can diminish the viability of human NSCLC (NCI-H460) cells with an IC_50_ of 50.82 μM. This IC_50_ value is higher than cisplatin (21.57 μM) as a positive control tested previously [30]. We first found that this phytochemical could also suppress the proliferation, migration, and anchorage-independent survival of NCI-H460 cells.

Another sister compound of cymbinodin-A (**2**), ephemeranthoquinone B (**3**), is firstly isolated from *Cymbidium* Great Flower Marie ‘Laurencin’ roots [21] and subsequently isolated from *Odontioda* Marie Noel ‘Velano’ [22] and *Cymbidium finlaysonianum* [18]. This phytochemical showed inhibitory activity against Gram-positive bacteria, human promyelocytic leukemia (HL-60) cells, human oral squamous cell carcinomas, and leukemic cells [21,22]. Masuda et al. [22] reported that anticancer activities of this compound were slightly higher than its cytotoxic effects on normal cell lines tested, and such cytotoxicity was not a result from apoptosis. *V*. *bensonii*-derived ephemeranthoquinone B (**3**) also showed mild cytotoxicity against NCI-H460 cells; however, dead cells confirmed by a nuclear staining method could be found when exposed to its cytotoxic doses. Additional investigations using high-resolution methods such as flow cytometry could be employed to discriminate modes of cell death (apoptosis and necrosis) caused by a tested molecule. Moreover, ephemeranthoquinone B (**3**) only affected NCI-H460 cell proliferation when exposed for 72 h (Figure 3) and did not inhibit migration and anchorage-independent growth of NCI-H460 cells (Figure 4 and Figure 5). The difference in anticancer activities between this compound and its sister molecule, cymbinodin-A (**2**), is highly likely to be due to the difference in their chemical structures. Ephemeranthoquinone B (**3**) consists of a double bond at C-9 and C-10 (Figure 1), whereas there is no double bond at this position for cymbinodin-A (**2**).

Protocatechuic acid (**4**) [23] seems to be a common phenolic compound found in diverse orchid species [24,25,26,27], and it is a notable intermediate in de novo biosynthesis of vanillin—the principal component of the vanilla orchid [24]. Our studies proved that protocatechuic acid (**4**) derived from V. bensonii showed no effect on the viability, proliferation, and migration of NCI-H460 cells (Figure 2, Figure 3 and Figure 4), but it exerted some roles to limit anchorage-independent growth of these cancer cells (Figure 5). Surprisingly, Tsao et al. [31] unveiled that this compound posed inhibitory activities against diverse NSCLC cells (i.e., A549, H3255, and Calu-6), while the underlying molecular mechanisms involved the modulation of FAK, MAPK, and NF-**κ**B pathways. The difference in these results relies significantly on the types of cancer cells tested. Another study demonstrated that this molecule from an edible mushroom, Clitocybe alexandri, could inhibit growth of NCI-H460 cells after treating for 48 h with an IC_50_ of 1616.0 *±* 75.3 μM [28]. Our findings agreed with this report as we could not see the inhibitory effect of V. bensonii-derived protocatechuic acid (**4**) on NCI-H460 cell viability, even when treating with the maximum tested dose of 100 μM for 24 h.

## 4. Materials and Methods

### 4.1. Plant Materials

*V*. *bensonii* plants were purchased in August 2011 from Chatuchak market in Bangkok, Thailand. Plant specimens were identified by Prof. Thatree Phadungcharoen (Department of Pharmacognosy and Pharmaceutical Botany, Faculty of Pharmaceutical Sciences, Chulalongkorn University) and deposited with a specimen voucher (BS-VB-082554) at the same institute.

### 4.2. Chemicals and Reagents

All organic solvents for the phytochemical extraction and elucidation (i.e., methanol (MeOH), ethyl acetate (EtOAc), dichloromethane (DCM), acetone, hexane, and dimethyl sulfoxide (DMSO)) and other chemical reagents (i.e., 3-(4,5-Dimethylthiazol 2-yl)-2,5-diphenyltetrazolium bromide (MTT), Hoechst 33342, propidium iodide (PI), crystal violet, formaldehyde, and agarose powder) were purchased from Sigma-Aldrich (St. Louis, MO, USA).

### 4.3. Extraction, Fractionation, and Structural Elucidation of Key Phytochemicals

The whole air-dried *V*. *bensonii* plants (1.9 kg) were cut, pulverized, and extracted with MeOH (3 × 5 L) at ambient temperature. The resultant organic solvent was evaporated under reduced pressure to give a dried mass of 221 g. The methanolic extract was partitioned with EtOAc and water to give an EtOAc extract (65.4 g) and an aqueous extract, respectively. The EtOAc extract was thereafter fractionated by vacuum liquid chromatography (VLC) on a silica gel (hexane–ethyl acetate, gradient) to give 5 fractions (I–V). Fraction III (15 g) was separated by column chromatography (CC, silica gel, EtOAc–DCM gradient) to yield 6 fractions (IIIa–IIIf). VLC and CC were performed on a silica gel 60 (Merck, Kieselgel 60, 70–320 µm) and a silica gel 60 (Merck, Kieselgel 60, 230–400 µm) (Darmstadt, Germany), respectively. Fraction IIId (26.2 mg) was further separated by CC (silica gel, hexane–acetone gradient) and then purified on a silica gel (hexane–MeOH gradient) to afford compound **1** (3.4 mg). Fraction V (24.5 mg) was separated on an EtOAc–DCM gradient to yield 9 fractions (Va–Vi). Thereafter, fraction Vh was further purified on a sephadex LH-20 column with acetone to obtain compound **2** (2.9 mg) and compound **3** (3.4 mg), respectively. Fraction Vi (8.0 mg) was fractionated by CC (silica gel, DCM–MeOH gradient) to obtain compound **4** (4.8 mg). Mass spectra were obtained on a Bruker micro TOF mass spectrometer (ESI-MS) (Billerica, MA, USA). NMR spectra were recorded on a Bruker Avance Neo 400 MHz NMR spectrometer (Billerica, MA, USA). 

### 4.4. Cell Culture and Growth Conditions

Human non-small cell lung cancer NCI-H460 cells were obtained from the American Type Culture Collection (ATCC) (Manassas, VA, USA). NCI-H460 cells were cultured in the Roswell Park Memorial Institute (RPMI) (Gibco, Gaithersburg, MA, USA). The culture medium was supplemented with 2 mM l-glutamine, 10% (*v*/*v*) fetal bovine serum, and 100 units mL^−1^ penicillin/streptomycin. The cells were cultivated at 37 °C under a 5% CO_2_ stream up to 70–80% confluence before use in the experiments.

### 4.5. Cell Viability Assay

The effects of *V*. *bensonii* metabolites on the viability of NCI-H460 cells were assessed using an MTT assay [32]. Cells seeded into a 96-well plate at a density of 8 *×* 10^3^ cells/well were exposed to varying concentrations of *V*. *bensonii* metabolites for 24 h. Cells treated with 0.5% (*v*/*v*) DMSO served as a vehicle control. The assay medium was removed and replaced with 0.5 mg mL^−1^ of MTT solution made by dissolving the compound with phosphate-buffered saline (PBS) and the incubation was continued in the dark at 37 °C under a 5% CO_2_ stream for 3 h. The assay solution was then removed and replaced with DMSO to solubilize the formazan crystal formed prior to the measurement of absorbance at the 570 nm wavelength by using a microplate reader (Anthros, Durham, NC, USA). The absorbance values derived were used to calculate the percentage of cell viability relative to the vehicle control. The experiments were performed at least in triplicate.

A dual Hoechst 33342/propidium iodide nuclear staining assay [33] was also conducted to confirm cell death after treating with varying concentrations of *V*. *bensonii* metabolites. Cells were seeded and treated as described above. The assay medium was removed and replaced with PBS containing 2 μg mL^−1^ Hoechst 33342 and 1 μg mL^−1^ propidium iodide. The assay plate was incubated in the dark at 37 °C under a 5% CO_2_ stream for 30 min. The stained cells were visualized and photographed under a fluorescence microscope (Olympus IX51 with DP70, Tokyo, Japan). Cells stained with bright blue (of Hoechst 33342) and red (of propidium iodide) were defined as dead cells. The experiments were performed at least in triplicate.

### 4.6. Cell Proliferation Assay

The impacts of *V*. *bensonii* metabolites on the proliferation of NCI-H460 cells were investigated using a crystal violet staining method [33]. NCI-H460 cells were seeded into a 96-well plate at a density of 2 × 10^3^ cells/well and treated with a non-cytotoxic concentration of *V*. *bensonii* metabolites. The assay plate was incubated at 37 °C under a 5% CO_2_ stream for 24, 48, and 72 h. When reaching each incubation time, the assay medium was removed and replaced with 0.05% (*v*/*v*) crystal violet solution containing 1% (*v*/*v*) formaldehyde and the incubation continued at ambient temperature for 30 min. The staining solution was removed, and stained cells were washed thrice with deionized water. Remaining stained cells were air-dried overnight and dissolved with MeOH before measuring the absorbance at the 570 nm wavelength using a microplate reader. The absorbance value corresponded to the number of proliferated cells in each treatment. The experiments were performed at least in triplicate, and cells treated with 0.5% (*v*/*v*) DMSO served as vehicle controls. 

### 4.7. Wound-Healing Assay

A wound-healing assay [34] was established to assess how *V*. *bensonii* metabolites affected NCI-H460 cell migration. Cells seeded into a 96-well plate at a density of 6 × 10^4^ cells/well were incubated at 37 °C under a 5% CO_2_ stream overnight. A wound made of a straight vertical line was formed by using a 20 μL pipette tip to scratch a cell monolayer at the center of each seeded well. The medium containing cell debris was removed and washed with PBS once. A non-cytotoxic concentration of each *V*. *bensonii* metabolite prepared by a dilution with RPMI medium was added to the assay well, while the one added to the medium containing 0.5% (*v*/*v*) DMSO served as a vehicle control. The settings were observed under a light microscope equipped with a camera (Nikon Ts2, Tokyo, Japan) and photographed to measure the wound area at 0, 12, and 24 h by using ImageJ software (https://imagej.nih.gov/ij/ accessed on 1 September 2022). The relative wound area to that measured at 0 h was computed, while the wound-healing rate referring to the cell migration rate was estimated from the slope of the linear regression plotted by the wound areas at 0 to 24 h. The experiments were performed at least in triplicate.

### 4.8. Soft Agar Colony Formation Assay

The capability of *V*. *bensonii* metabolites to inhibit survival and growth of NCI-H460 cells under anchorage-independent conditions was investigated using a soft agar colony formation assay [35]. The soft agar with two layers was constructed in a 24-well plate, where bottom and top layers contained 0.5% and 0.3% (*w*/*v*) agarose (250 μL each), respectively. Both agar layers were prepared by diluting 1.5% (*w*/*v*) agarose with RPMI medium. The bottom layer was established up to 4 h prior to the top layer, which contained NCI-H460 cells at a density of 1.5 × 10^3^ cells/well and each *V*. *bensonii* metabolite at a non-cytotoxic concentration or 0.5% (*v*/*v*) DMSO as a vehicle control. When the top agar layer was set, 250 μL of RPMI medium was added. The settings were incubated at 37 °C under a 5% CO_2_ stream for one week with the addition of the medium every 3 days. When reaching the incubation time, NCI-H460 colonies formed in the agar were observed, counted, and photographed under the light microscope equipped with a camera. Derived photographs were used to measure the colony size with the ImageJ software. Colony number and size referred to the survival and growth capabilities of NCI-H460 cells under anchorage-independent conditions.

### 4.9. Statistical Analysis

Numerical results were presented as means ± standard deviations or standard errors and relative values to the control. The statistical comparison of means was conducted with one-way analysis of variance at *⍺* = 0.05, while a pairwise comparison of each mean to that of the control was processed with Tukey’s post hoc test. The statistical significance (*p*) of the difference in a pairwise comparison of means was determined as ns = no significance (*p* > 0.05), * = *p* < 0.05, ** = *p* < 0.01, *** = *p* < 0.0005, and **** = *p* < 0.0001. All statistical analyses were conducted with GraphPad Prism 8.0.2 software (San Diego, CA, USA).

## 5. Conclusions

Four phytochemicals, including phloretic acid methyl ester (**1**), cymbinodin-A (**2**), ephemeranthoquinone B (**3**), and protocatechuic acid (**4**), were isolated from the whole plant methanolic extract of *V*. *bensonii*. These compounds and the crude methanolic extract demonstrated varying bioactivities to inhibit growth and metastasis of NCI-H460 cells. Overall, the crude extract and phloretic acid methyl ester (**1**) offered the greatest bioactivities to limit the viability, proliferation, migration, and anchorage-independent survival and growth of these lung cancer cells. To this end, it is proved that *V*. *bensonii* is a promising source of bioactive molecules possessing anticancer potential, which can be developed further as pharmaceutical products to optimize the future pharmacotherapy of lung cancer. Additional studies to unveil the molecular mechanisms underlying such anticancer activities of these *V*. *bensonii* metabolites and in vivo assessments are essential to fill the knowledge gaps in their pharmaceutical applications.

## Figures and Tables

**Figure 1 molecules-27-07902-f001:**
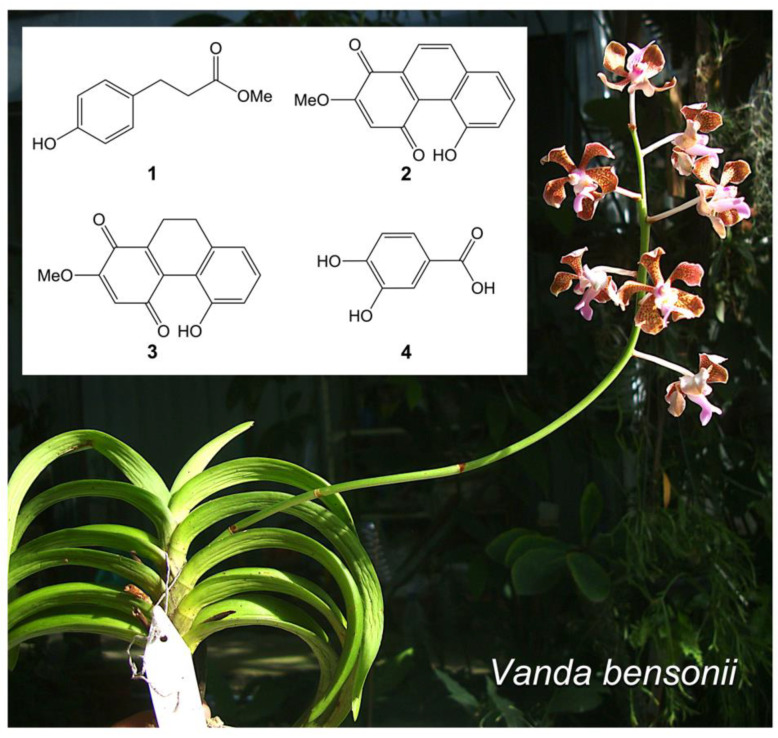
*Vanda bensonii* and its specialized metabolites. Phytochemicals isolated from the methanolic extract of this orchid plant include phloretic acid methyl ester (**1**), cymbinodin-A (**2**), ephemeranthoquinone B (**3**), and protocatechuic acid (**4**).

**Figure 2 molecules-27-07902-f002:**
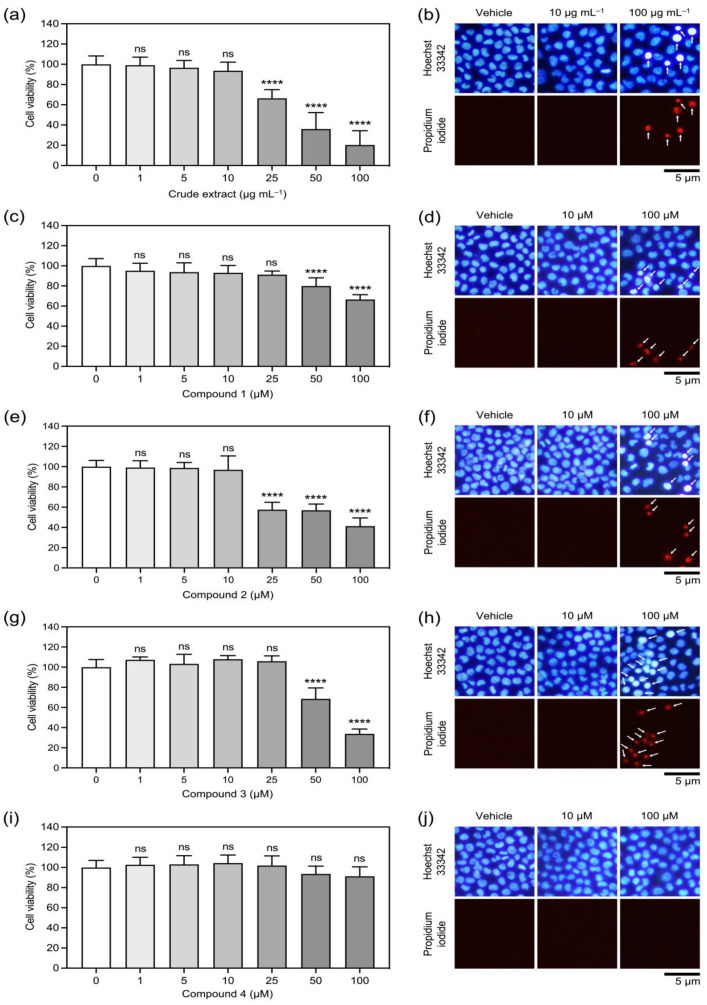
Cytotoxic effects of *Vanda bensonii* metabolites on the viability of NCI-H460 cells. Percentages of cell viability after treating cells with varying concentrations of *V*. *bensonii* metabolites for 24 h, assessed by a colorimetric method (**a**,**c**,**e**,**g**,**i**) and confirmed by a dual Hoechst 33342/propidium iodide nuclear staining assay (**b**,**d**,**f**,**h**,**j**). Tested compounds comprise of a crude extract (**a**,**b**), phloretic acid methyl ester (**1**) (**c**,**d**), cymbinodin-A (**2**) (**e**,**f**), ephemeranthoquinone B (**3**) (**g**,**h**), and protocatechuic acid (**4**) (**i**,**j**). All experiments were performed at least in triplicate, and 0.5% (*v*/*v*) dimethyl sulfoxide served as the vehicle control. The graphical results represent means ± standard deviations, and statistical comparisons of means between each treatment and the control were conducted at *⍺* = 0.05, where ns = no significance and **** = *p* < 0.0001. Representative images derived from the dual nuclear staining were randomly selected, where the arrows pointed out dead cells.

**Figure 3 molecules-27-07902-f003:**
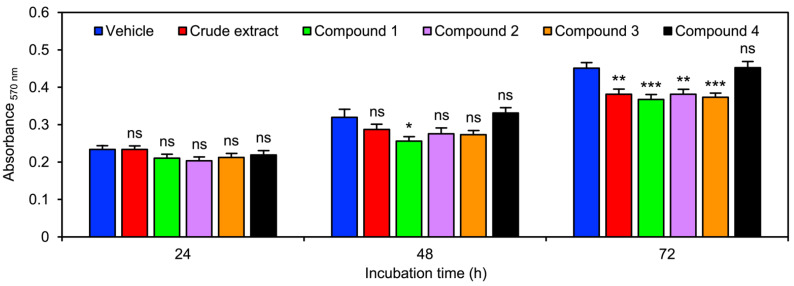
Impacts of *Vanda bensonii* metabolites on NCI-H460 cell proliferation. The capability of cells to proliferate in the presence of each tested compound from *V*. *bensonii* (i.e., a crude extract, phloretic acid methyl ester (**1**), cymbinodin-A (**2**), ephemeranthoquinone B (**3**), and protocatechuic acid (**4**)) at 10 μg mL^−1^ (for the crude extract) or 10 μM (for the isolated compounds) for different incubation durations was assessed by a colorimetric method. The result was reported as the absorbance value at a 570 nm wavelength that correlated to the density of cells measured from each treatment. All experiments were performed at least in triplicate, and 0.5% (*v*/*v*) dimethyl sulfoxide served as the vehicle control. The graphical results represent means ± standard errors of means, and statistical comparisons of means between each treatment and the control were conducted at *⍺* = 0.05, where ns = no significance, * = *p* < 0.05, ** = *p* < 0.01, and *** = *p* < 0.0005.

**Figure 4 molecules-27-07902-f004:**
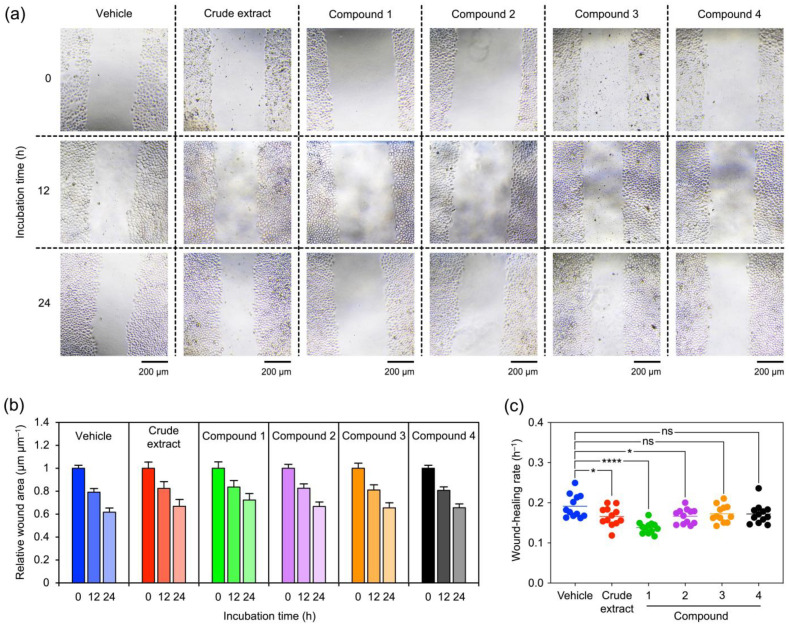
Impacts of *Vanda bensonii* metabolites on NCI-H460 cell migration. The capability of H460 cells to migrate in the presence of each tested compound (i.e., a crude extract, phloretic acid methyl ester (**1**), cymbinodin-A (**2**), ephemeranthoquinone B (**3**), and protocatechuic acid (**4**)) at 10 μg mL^−1^ (for the crude extract) or 10 μM (for the isolated compounds) for different incubation durations was assessed by a wound-healing assay. (**a**) Representative images show the change in wound areas measured at different incubation times. (**b**) The change in wound areas was calculated and reported as the relative values to those detected at 0 h. (**c**) The wound-healing rate for each tested compound was also computed. All experiments were performed at least in triplicate, and 0.5% (*v*/*v*) dimethyl sulfoxide served as the vehicle control. The graphical results represent means ± standard errors, and statistical comparisons of means between each treatment and the control were conducted at *⍺* = 0.05, where ns = no significance, * = *p* < 0.05, and **** = *p* < 0.0001.

**Figure 5 molecules-27-07902-f005:**
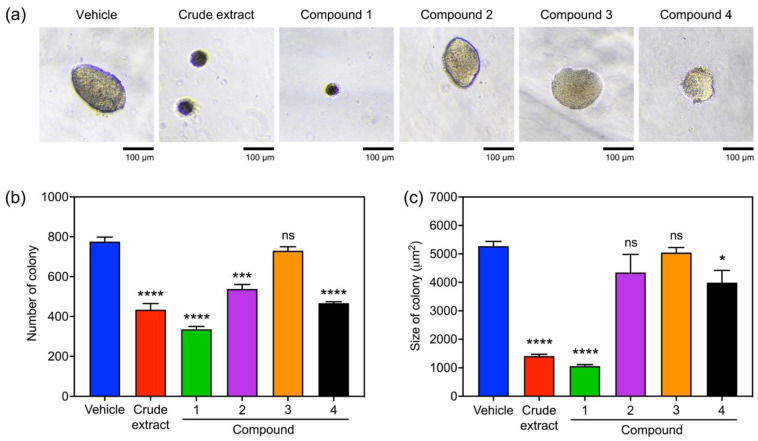
Impacts of *Vanda bensonii* metabolites on the anchorage-independent growth of NCI-H460 cells. The capability of H460 cells to grow without anchorage in the presence of each *V*. *bensonii* metabolite (i.e., a crude extract, phloretic acid methyl ester (**1)**, cymbinodin-A (**2**), ephemeranthoquinone B (**3**), and protocatechuic acid (**4**)) at 10 μg mL^−1^ (for the crude extract) or 10 μM (for the isolated compounds) for 7 days was assessed by a soft agar colony formation assay. (**a**) Representative images show varying sizes of H460 colonies formed. The differences in numbers (**b**) and sizes (**c**) of H460 colonies were computed. All experiments were performed at least in triplicate, and 0.5% (*v*/*v*) dimethyl sulfoxide served as the vehicle control. The graphical results represent means ± standard errors, and statistical comparisons of means between each treatment and the control were conducted at *⍺* = 0.05, where ns = no significance, * = *p* < 0.05, *** = *p* < 0.0005, and **** = *p* < 0.0001.

**Table 1 molecules-27-07902-t001:** Some bioactivities of phytochemicals derived from *Vanda bensonii*.

Phytochemicals from *V*. *bensonii*	Absolute IC_50_ ^1^	Fold Change in Cell Proliferation ^2^		Wound-Healing Rate (h^−1^) ^3^	Anchorage Independence ^4^	
		48 h	72 h		Fold Change in Survival	Fold Change in Growth
Crude methanolic extract	40.39 μg mL^−1^	0.90	0.85	0.16 ± 0.02	0.56	0.27
Phloretic acid methyl ester (**1**)	>100 μM	0.80	0.81	0.14 ± 0.01	0.43	0.20
Cymbinodin-A (**2**)	50.82 μM	0.86	0.85	0.17 ± 0.02	0.69	0.82
Ephemeranthoquinone B (**3**)	64.22 μM	0.85	0.83	0.17 ± 0.02	0.94	0.96
Protocatechuic acid (**4**)	>100 μM	1.04	1.00	0.17 ± 0.02	0.60	0.76

^1^ Cytotoxic effect of each *V*. *bensonii* phytochemical on NCI-H460 cells was reported as the absolute half maximal inhibitory concentration (absolute IC_50_) value derived from the sigmoidal dose-response plot (Appendix A). ^2^ NCI-H460 cell proliferation after treating with 10 μg mL^−1^ or 10 μM of each *V*. *bensonii* phytochemical for 48 and 72 h was assessed using crystal violet staining assay, while the fold change of proliferated cells from those of the vehicle control made of 0.5% (*v*/*v*) dimethyl sulfoxide (DMSO) was calculated. ^3^ The wound-healing rate of NCI-H460 cells after treating with each *V*. *bensonii* phytochemical was computed based on linear regressions of reduced wound areas across times of detection, which represents the cell migration rate derived from three independent experiments—the results refer to means ± standard deviations. ^4^ The capability of NCI-H460 cells to survive and grow in the presence of 10 μg mL^−1^ or 10 μM of each *V*. *bensonii* phytochemical under anchorage-independent conditions was evaluated using the soft agar colony formation assay. After treating for 7 days, the fold changes in survival and growth of NCI-H460 colonies from those of the vehicle control made of 0.5% (*v*/*v*) DMSO were accounted based on the colony counts and sizes, respectively.

## Data Availability

Not applicable.

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
