# Peer review of "Phytochemicals from Vanda bensonii and Their Bioactivities to Inhibit Growth and Metastasis of Non-Small Cell Lung Cancer Cells"

_molecules, 2022, doi:10.3390/molecules27227902_

Round 1
Reviewer 1 Report
Article
Natural Products from Vanda bensonii and Their Bioactivities to
Inhibit Growth and Metastasis of Non-Small Cell Lung Cancer Cells
Non-small cell lung cancer (NSCLC) is becoming very prevalent.
In this manuscript, authors have studied a type of notable Vanda species (Orchid) from Thailand,
Vanda bensonii Batem is an economically important orchid species due to its unique features.
Vanda species is still unknown for its chemical constituents and their pharmaceutical potential.
Here authors have elucidated their capability to inhibit the growth and metastasis of NSCLC using the human
NCI-H460 cells as a study model, the NSCLC cells were chosen based on their origin from pleural fluid,
which represent high metastatic features including the capability to grow under anchorage independence conditions.
The manuscript is well structured, with lots of data
adequate figures and tables
The literature review seems incomplete, may revise it accordingly.
Very few grammatical mistakes were there, which can be removed.
Conclusions can be improved.
Accept the article for publication in molecules after minor mandatory revisions.
With Regards,
Author Response
Article
Natural Products from Vanda bensonii and Their Bioactivities to Inhibit Growth and Metastasis of Non-Small Cell Lung Cancer Cells
Non-small cell lung cancer (NSCLC) is becoming very prevalent.
In this manuscript, authors have studied a type of notable Vanda species (Orchid) from Thailand, Vanda bensonii Batem is an economically important orchid species due to its unique features. Vanda species is still unknown for its chemical constituents and their pharmaceutical potential.
Here authors have elucidated their capability to inhibit the growth and metastasis of NSCLC using the human NCI-H460 cells as a study model, the NSCLC cells were chosen based on their origin from pleural fluid, which represent high metastatic features including the capability to grow under anchorage independence conditions.
The manuscript is well structured, with lots of data adequate figures and tables
The literature review seems incomplete, may revise it accordingly.
Very few grammatical mistakes were there, which can be removed.
Conclusions can be improved.
Accept the article for publication in molecules after minor mandatory revisions.
With Regards,
Response: Thank you very much for your evaluation and comments. We re-checked and improved our manuscript with a thorough proofread and valid English usage where appropriate. We improved the sense and flow of the introduction and added some additional background relevant to our work. The conclusions were amended to cover our goals and what we found.
Reviewer 2 Report
I appreciate the authors for their excellent work. I have a few suggestions:
1. Modify the title of the manuscript in the place of 'Natural Products' in the title add 'Phytochemicals'. Natural Products term used in a wide spectrum. If we use Phytochemicals it means related to the plant used in the studies.
2. In the result section in subheading 2.1 line number 130 in the place of 'orchid plant' add the name of the orchid 'V. bensonii'
Author Response
I appreciate the authors for their excellent work. I have a few suggestions:
1. Modify the title of the manuscript in the place of 'Natural Products' in the title add 'Phytochemicals'. Natural Products term used in a wide spectrum. If we use Phytochemicals it means related to the plant used in the studies.
2. In the result section in subheading 2.1 line number 130 in the place of 'orchid plant' add the name of the orchid ' bensonii'
Response: Thank you very much for your evaluation and comments. We revised the title and subheading 2.1 accordingly.
Reviewer 3 Report
- In Table 1, cytotoxicity values should be definite values. Please specify values that are > 10, 25, 50, and 100.
- Please add sigmodal dose-resposce curve for the IC50 calculation
- In figure for the PI staining, please add arrows.
- What is the significance of figure 3 to calculate the Absorbance compared to the incubation time
- What about to calculate the Apoptosis/necrosis assemenet using flow cytometry
-Please revise the statistical significance (**)+(***), Please be careful with the statistical signs
- Please, where are the NMR analysis for the characterized compounds
Author Response
Response: Thank you very much for your evaluation and comments.
- In Table 1, cytotoxicity values should be definite values. Please specify values that are > 10, 25, 50, and 100.
Response: We replaced these data with absolute IC50 values.
- Please add sigmodal dose-resposce curve for the IC50 calculation
Response: In connection with the previous comment, we included the sigmoidal plots in the “Supplementary Materials” file as Figure S1 (lines 712-714) and updated the abstract and text accordingly.
- In figure for the PI staining, please add arrows.
Response: We added the arrows to point out the dead cells in both Hoechst 33342- and PI-stained fields with descriptions in the Figure’s caption (line 355) and in text (lines 640-642 and 646-647).
- What is the significance of figure 3 to calculate the Absorbance compared to the incubation time
Response: The correlation between absorbance value and cell density was described in the Figure’s caption (lines 384-386) and text (lines 659-660).
- What about to calculate the Apoptosis/necrosis assemenet using flow cytometry
Response: Flow cytometry will offer more precise outcomes to discriminate modes of cell death. However, a dual Hoechst 33342/PI nuclear staining assay is valid and sufficient to visualize and confirm the presence of dead cells caused by different treatments and controls. Based on our preliminary study, we wish to convince the readers that the tested phytochemicals can pose cytotoxic effects on NSCLC cells, and cell death can be a consequence of such effects. We mentioned and described about this staining method in lines 640-642 and 646-647 and its limitation in lines 550-552.
- Please revise the statistical significance (**)+(***), Please be careful with the statistical signs
Response: We carefully checked the statistical results and interpretations again, and they are presented correctly. Could the reviewer specify which error we should revise? The statistical analysis was described and reported in lines 696-699 and all relevant Figures’ captions. We conducted the statistical analysis at ⍺ = 0.05, and the statistical significance (p) of the difference in a pairwise comparison of means was determined as ns = no significance (p > 0.05), * = p < 0.05, ** = p < 0.01, *** = p < 0.0005, and **** = p < 0.0001. We included this additional explanation in section 4.9 Statistical analysis (lines 697-699).
- Please, where are the NMR analysis for the characterized compounds
Response: We described NMR data for each compound in section 2.1 of the Results and section 4.3 of the Materials and Methods. Due to these phytochemicals are known and have previous NMR-based elucidation data, we, therefore, omit their NMR discussion to avoid redundancy in data presentation.
Round 2
Reviewer 3 Report
The authors have addressed all comments raised by reviewers, so it may be accepted at its current status